# Design of a pH-Responsive Conductive Nanocomposite Based on MWCNTs Stabilized in Water by Amphiphilic Block Copolymers

**DOI:** 10.3390/nano9101410

**Published:** 2019-10-03

**Authors:** Frank den Hoed, Andrea Pucci, Francesco Picchioni, Patrizio Raffa

**Affiliations:** 1Department of Chemical Engineering—Product Technology, University of Groningen, Nijenborgh 4, 9747 AG Groningen, The Netherlands; f.m.den.hoed@rug.nl (F.d.H.); f.picchioni@rug.nl (F.P.); 2Department of Chemistry and Industrial Chemistry, University of Pisa, Via Giuseppe Moruzzi 13, 56124 Pisa (PI), Italy; andrea.pucci@unipi.it

**Keywords:** amphiphilic block copolymers, carbon nanotubes, stimuli responsive, conductive composite

## Abstract

Homogeneous water dispersions of multi-walled carbon nanotubes (MWCNTs) were prepared by ultrasonication in the presence of an amphiphilic polystyrene-block-poly(acrylic acid) (PS-b-PAA) copolymer. The ability of PS-b-PAA to disperse and stabilize MWCTNs was investigated by UV-vis, SEM and zeta potential. The results show that the addition of a styrene block to PAA enhances the dispersion efficiency of the graphitic filler compared to pure PAA, possibly due to the nanotube affinity with the polystyrene moiety. Notably, the dispersions show an evident pH-responsive behavior, being MWCNTs reaggregation promoted in basic environment. It is worth noting that the responsive character is maintained in solid composites obtained by drop casting, thus indicating potential applications in sensing.

## 1. Introduction

Carbon nanotubes (CNTs) are tube-shaped allotropes of carbon with a diameter of a few nanometers. They have received great interest since their discovery in 1991; because of their remarkable properties [1,2], in particular their mechanical strength; thermal properties and electrical conductivity [3,4,5]. CNTs may be used in applications like transistors; batteries; conductive films; sensors [6,7] and for mechanical reinforcement in composite materials [8,9,10,11,12,13,14].

Due to the high aspect ratio of CNTs, they have great van der Waals (vdW) attracting forces keeping the nanotubes in an aggregated state [15] thus severely limiting possibilities to explore their full potential. CNTs dispersion in water can be of particular interest for biological applications [16] even if strongly affected by their hydrophobic nature that limits the dispersion stability over time. Notably, CNTs can form stable dispersions in some solvents with similar surface tension like NMP and DMF [17,18] even if only dilute dispersion can be made by this method. Conversely, chemical functionalization of CNTs enables the preparation of dispersions at higher concentration, and provides the possibility to attach groups with affinity for a desired dispersing medium. To make room for covalent bonds, defects on the CNT walls need to be created. This requires harsh conditions, such as high temperatures combined with highly reactive chemicals [19,20,21]. Disadvantages of covalent functionalization are the damaging of nanotubes during the process, with subsequent loss of favorable CNT properties, and the use of environmentally unfriendly chemicals. Alternatively, non-covalent functionalization of nanotubes is accessible as well. By this method, the surface of the nanotubes is kept intact, i.e., preserving the electrical conductivity [22]. It is worth noting that various classes of substances can be used for non-covalent stabilization of CNTs in water. A possible approach is the use of surfactants like, for example, sodium dodecyl sulfate (SDS) and related salts that are known to physically adsorb to the nanotube surface [23]. The hydrophobic part of a surfactant has good affinity for the nanotube wall, while the hydrophilic head promotes the dispersion in water. Affinity is optimized when the surfactant contains π-conjugated moieties such as aromatic amines, styrene or pyrene [17,24,25,26] that can effectively interact with the nanotubes via π-stacking. Moreover, the use of charged surfactants can more efficiently prevent re-aggregation of nanotubes by electrostatic repulsion. Another type of substance for nanotube stabilization are water soluble polymers. The use of polymers enables the activation of steric hindrance as an entropic (thermodynamic) stabilization mechanism in addition to the electrostatic (kinetic) repulsion [17]. The affinity for the nanotubes by water soluble polymers can be achieved by its hydrophobic backbone and rendered even more effective due to the presence of π-conjugated moieties in the polymer structure (Figure 1a). Such a polymer has multiple points of interaction with the nanotubes resulting in static dispersion, which is different from the dynamic dispersion mechanism provided by surfactants (Figure 1b–d). Surfactants are more easily removed, by for example centrifugation or filtration, because of the dynamic nature of the dispersion [27,28].

It is further reported that when the substance used for stabilization is responsive to external stimuli, this enables the possibility of making a smart material with sensor abilities [29]. Polyelectrolytes for example are water soluble polymers responsive to pH and salinity due to their large number of ionizable groups. [30] Poly(acrylic acid) (PAA) has been shown to effectively disperse nanotubes in water [31,32] depending on pH. At high pH (>8) PAA contains many charged groups, which significantly decrease the affinity for CNT and thus decreases the CNT dispersion abilities [32,33,34]. The affinity could be improved by attaching a π-conjugated moiety, like polystyrene (PS), to the PAA chain as this allows for π-π non-covalent bonds. A copolymer of PAA and polystyrene should therefore result in more stable dispersions (Figure 1), which could also display potential sensitive features to external stimuli such as pH and salinity [35,36,37,38]. Actually, the dispersion stability compared to PAA could be significantly improved by micelle formation around the CNT, as it has been shown in water/DMF solution for cross-linked PS-b-PAA by Kang and Taton [18]. In the mentioned study, the stabilization of CNTs in solution is demonstrated, but no composites were prepared and no investigation of the responsive properties has been performed. Better control and higher stability in aqueous CNT dispersions combined with responsive properties are of vital importance to explore the full potential of CNTs and use them as smart materials.

Herein, we propose to study the preparation of multi-walled CNTs (MWCNTs) dispersions directly stabilized in water by means of several PS-b-PAA copolymers with variable length of the PAA block. The PAA length is varied because it governs conformational changes with pH variations [39], and it is maintained several times longer than the PS chain to provide the copolymers water solubility. The copolymers were prepared by Atom Transfer Radical Polymerization (ATRP) [40], according to previously published procedure [36]. ATRP allowed for precise design of the polymer, control over block length and narrow molecular weight distribution. First, a PS macroinitiator was synthesized and chain extended with tert-butyl acrylate, which was eventually hydrolyzed to yield the PAA block. The two-step approach for the attachment of the PAA chain is required because polymerization control is poor for copper-mediated ATRP in protic environments [41].

MWCNTs/PS-b-PAA copolymer dispersions were then prepared by ultrasonication and their pH-responsive behavior studied by spectroscopy and microscopy investigations. Furthermore, solid composites realized by drop casting were studied in terms of their electrical-responsiveness towards pH variations.

## 2. Materials and Methods

### 2.1. Materials

Multi-walled carbon nanotubes (CNTs) were purchased from Sigma-Aldrich (St. Louis, MO, USA) (product number: 791431, lot MKBT4011V) and used as received. Methyl 2-bromopropionate (MBP, Sigma-Aldrich (St. Louis, MO, USA), ≥99%), styrene (Sigma-Aldrich (St. Louis, MO, USA), ≥99%), *N*,*N*,*N*′,*N*′′,*N*′′-pentamethyldiethylenetriamine (PMDETA, Sigma-Aldrich(St. Louis, MO, USA), 99%), anisole (Sigma-Aldrich (St. Louis, MO, USA), 99.7%), glacial acetic acid, methanol, dioxane, tetrahydrofuran (THF), dimethylformamide (DMF), ethanol, ethyl acetate and 1,4-diazabicyclo[2.2.2]octane (Sigma-Aldrich, analytical grade) were used without any purification. Tert-butyl acrylate (Sigma-Aldrich (St. Louis, MO, USA), 98%) was purified over a column of basic aluminum oxide and stored under nitrogen before use. Copper(I)bromide (Sigma-Aldrich (St. Louis, MO, USA), 99%) and copper(I)chloride (Sigma-Aldrich (St. Louis, MO, USA), 99%) were stirred in glacial acetic acid for 5 h, filtered, washed with acetic acid, ethanol and ethyl acetate and dried under vacuum before use.

### 2.2. Polymers Preparation

#### 2.2.1. Preparation of PS-Br Macroinitiator

The styrene macroinitiator (PS-Br) was prepared by the following procedure, in agreement with our previously reported research [36]: 5–10 mmol MBP, 5 mmol Cu(I)Br and 300 mmol styrene was dissolved in 20 mL anisole in a 250 mL three-necked round bottom flask. The flask was placed in an oil bath at 100 °C. Air was removed by bubbling nitrogen gas through the solution for at least 30 min. Then, 10 mmol of PMDETA were added to start the reaction. After 5 h, the reaction was stopped by cooling down to room temperature, and 50–100 mL of THF was added under air atmosphere. The copper catalyst was removed by filtration over a neutral alumina column. The crude product was precipitated in an excess of methanol, filtered, redissolved in THF, reprecipitated in methanol:water (2:1 *v*/*v*), washed with methanol and dried overnight at 60 °C. A white solid was obtained. The molecular weight of the polymer was determined by NMR and gel permeation chromatography (GPC).

#### 2.2.2. Preparation of PS-b-PAA Polymer

To prepare the second block (ptBA), 1 g of macroinitiator (PS-Br) was dissolved in 15 mL anisole under nitrogen. Cu(I)Cl and monomer (tBA) were added according to the desired stoichiometry. The flask was placed in an oil bath at 90 °C and nitrogen was bubbled through the solution for at least half an hour before PMDETA was added. The reaction was stopped after a given time by cooling down to rt, and 50 mL of THF was added under air atmosphere. The copper catalyst was removed by filtration over a neutral alumina column. The crude product was precipitated in an excess of methanol:water (2:1 *v*/*v*), filtered, redissolved in THF, reprecipitated in methanol:water (2:1), washed with methanol and dried overnight at 60 °C. A white solid was obtained. The molecular weight of the polymer was determined by NMR and GPC.

The resulting polymers were hydrolyzed in a 1,4-dioxane solution in a round-bottom flask. Approximately 20 mL of dioxane per gram of polymer were used. The flask was equipped with a stirring egg, a reflux condenser and was heated to 100 °C in an oil bath. After an hour, HCl was added (2 mL more than stoichiometrically required). The reaction was stopped after 3 h by cooling down. The solution was precipitated in acetone, filtered and dried at 60 °C. The extent of hydrolysis was determined by NMR in DMSO-d6.

### 2.3. Nanocomposite Preparation and Setup Preparation for the Resistive and pH-Responsive Behaviour

About 15 mg of PS-b-PAA were dissolved in 3 mL of water in a 20 mL vial. Then 1 mg of MWCNTs was poured into the vial and ultrasonicated for 5 min with a probe sonicator model UP 400 S by Hielscher Ultrasound at 60% of power 0.5 s^−1^ frequency. An ice bath was used to prevent solvent evaporation during sonication. The dispersion was then diluted with water to a polymer concentration of 0.46 mg/mL and (if pH was adjusted) a 1 M NaOH solution was added dropwise until the desired pH was measured. The dispersion was sonicated for another 3 min, centrifuged at 3000 rpm and filtered before being characterized. For samples with different nanotube concentrations, the amount of nanotubes was varied whereas the polymer concentration and liquid volume was kept constant.

For the determination of the resistive behavior in the solid state, 25 µL of each water dispersion was drop casted onto gold electrodes supported on an integrated device provided by Cad Line Pisa (Italy). The electrodes were fabricated onto FR-4 that is a composite material composed of woven fiberglass cloth with an epoxy resin binder substrate (thickness of 2 mm). Copper tracks were obtained by photolithography and electroplated with nickel and gold to fabricate the electrodes (thickness of copper 35 μm, nickel 3.0 μm, and gold 1.2 μm). After complete water evaporation by drying the devices in an oven at 120 °C for 5 min, the electrical resistance was measured using a Keithley 2000 multimeter as a mean from one hundred measures according to the multimeter settings.

For the determination of the pH responsive behavior, the electrodes with drop casted films were submerged in an acetone solution containing 5 g/L of 1,4-diazabicyclo [2.2.2]octane (DABCO, pKa = 8.82) [42]. The electrodes were then removed from the DABCO solution after 15 min and dried at room temperature for 24 h. Subsequently, the electrical resistance was again measured with the same setup.

### 2.4. Characterization

Gel permeation chromatography (GPC) measurements were carried out with a HP1100 machine (Agilent Technologies, Waldbronn, Germany) equipped with one guard column (PL-gel 5 μm Guard, 50 mm) followed by two columns of PL-gel (5 μm Mixed-C, 300 mm) in series, and detection system based on refractive index GBC LC 1240 (GBC Scientific Equipment Pty Ltd., Victoria, Australia). The samples (5 mg/mL of polymer in THF plus 1 drop of toluene as internal standard) were eluted with THF at a rate of 1 mL/min, at 140 bar of pressure and 40 °C. Molecular weights and polydispersity index (PDI) were determined using the software PSSWinGPC Unity from the Polymer Standard Service. Polystyrene standards were used for calibration.

To determine the molecular weight of the ptBA homopolymer, a triple detection system was used, based on: a Viscotek Rals detector, a viscometer H502 and a shodex RI-71 refractive index detector. A dn/dc value of 0.0479 mL/g was used for the ptBA chains.

UV-Vis absorbance spectra of the prepared dispersion were recorded at room temperature with a PerkinElmer Lambda 650 (Waltham, MA, USA) spectrometer from wavelength 300 to 600 nm and using 1 cm cuvette.

The zeta potential was measured by Brookhaven ZetaPALS (Holtsville, NY, USA). Ten cycles were performed per each sample. A polymer concentration of 6 mg per mL MilliQ water was used. For samples containing MWCNTs, a feed of 0.22 mg/mL of nanotubes was used.

The morphology of the solid dispersions was investigated by FEI Quanta 450 FEG Environmental Scanning Electron Microscope (SEM) pictures (ThermoFisher scientific, Hillsboro, OR, USA). The MWCNTs/polymer samples for SEM were ultrasonically dispersed in water for analysis. The suspensions were deposited on a gold-coated silicon wafer and allowed to dry in a vacuum system overnight. The wafer was then mounted onto a stainless steel sample holder using carbon tape.

Thermal degradation of the polymers and nanocomposites were analyzed via thermogravimetric analysis (TGA) with a Mettler Toledo TGA/SDTA851 instrument (Columbus, OH, USA) under nitrogen flux. All samples were tested in the temperature range of 25 °C to 450 °C with a scan rate of 10 °C/min.

## 3. Results and Discussion

### 3.1. Polymers Synthesis

Several polymers were designed and synthesized by the procedure shown in Figure 2, based on our previous research [36]. Conditions are described in Appendix A and NMR characterization is reported in Appendix A. The length of the blocks is expressed in the sample name. For example, PS_26_PAA_81_ is a diblock copolymer consisting of a polystyrene chain of 26 units and a polyacrylic acid chain of 81 units (approximately). Details of synthesis and characterization can be found in the Appendix A. The relatively short hydrophobic block (26 units) combined with a long hydrophilic block allowed for the polymers water solubility.

### 3.2. Polymer/MWCNT Dispersions in Water: Effect of Polymer Structure on Stability

The copolymers were dissolved in water (0.46 mg/mL) and the MWCNTs dispersed by ultrasonication for 5 min (0.03 mg/mL feed) [43,44]. After possible pH-adjustment, samples were sonicated for another 3 min. To estimate the amount of MWCNTs effectively stabilized by the prepared copolymers at different pH, UV-Vis spectra of the dispersion were recorded after centrifugation (Appendix A). The amount of light absorbed or scattered by dispersion is correlated to the MWCNTs concentration [34]. The intensity at a given arbitrary wavelength (450 nm) would be proportional to the amount of MWCNTs present in the dispersion, according to the Lambert-Beer law. In Figure 3, the light absorption at 450 nm is shown for different polymer/MWCNTs dispersions in water at various pHs.

Since all dispersions contain the same copolymer mass, the shorter chains have a higher molar concentration. On the other hand, longer chains provides higher surface coverage that can be estimated from the radius of gyration of the PAA chain, assuming that the polymers adsorb on the CNTs surface as single chains (Figure 4). A mathematical derivation of the total surface coverage is reported in the Appendix A. The calculated total surface coverage for each sample is given in Table 1.

Without any pH adjustment, the pH of the dispersion was 5 for all polymers. In this condition, the PAA homopolymer had the lowest dispersion efficiency (Figure 3). This shows that the highest surface coverage (see Table 1) did not correspond to a better CNT dispersion. The polymer giving the best dispersion (PS_26_PAA_81_) had the highest relative polystyrene content, which should contribute to a more effective π–π interaction with the CNTs. This suggests that the overall CNT/polymer affinity was more limiting than the stabilizing effects provided by the PAA chains. The assumption of single chain adsorption in the spherical form used for the model in Table 1 and Figure 4 was therefore too simplistic.

The polymers used in this study were known to form colloidal micellar aggregates in water, as we have discussed in previous research [35,36,37,38]. If the polymers would form micelles around the MWCNTs like surfactants do [45] (as is visualized in Figure 5), the PAA length would not make a difference as it merely determines the thickness of the protecting layer between the nanotube and water. As long as this is thick enough, the molar concentration of the polymer would be determining the amount of nanotubes that can be dispersed. Based on these considerations, we could suggest that in our system, the sonication partially disrupted the polymeric micellar aggregates and caused a rearrangement of polymeric chains around the CNTs, based on their affinity for the polystyrene block. The hypothesis that sonication could disrupt the micellar aggregates in solution, although not well documented in literature to the best of our knowledge, it is supported by the experimental observation that the viscosity of water solutions of PS-b-PAA, initially high, decreased significantly upon sonication.

### 3.3. Effect of pH on Stability of Polymer/MWCNT Dispersions

Another factor to take into account is that the PAA chain conformation is also dependent on its degree of protonation, therefore on the solution pH. At low pH, PAA is mostly present in coiled formation [46,47,48] and shorter chains only have a slightly smaller radius compared to the longer ones. Therefore, they can cover a larger amount of the nanotube surface at the same polymer weight concentration (higher molar concentration).

Moreover, while the π–π interaction of the PS block did not cause any strain on the chain, the slight deprotonation of PAA between pH 5 and 7, caused a more stretched conformation, and consequently a more limited contact with the nanotube. The weakly ionized polymer can non-covalently bind to the nanotube surface [32].

As the pK_a_ of acrylic acid is 4.25 and approximately 4.5 for polyacrylic acid [39], at pH higher than 5 the PAA chain in the polymer becomes deprotonated. This causes intramolecular repulsions between charged units, resulting in a transition from coil to elongated chain and an increase of the radius of gyration [47]. If a micellar structure were formed around the nanotubes, this would increase the thickness of the PAA layer and improve stabilization, which would contradict what was observed in Figure 3. The results show in fact that less nanotubes were dispersed at higher pH, especially for the shortest PAA chains, thus suggesting that the stabilization had become weaker. Therefore, this behavior contradicted the hypothesis of a micellar structure around the nanotubes.

Erika et al. suggested that pure PAA in water stabilizes CNT by forming globular structures parallel to the surface of the nanotubes [32,33,34]. The mentioned research also found a decrease in nanotubes dispersed at high pH as the conformation of the polymer changes from globular to stretched (still parallel to the nanotube surface). In contrast with a micelle model, where stretching of the polymer led to increased steric hindrance (Figure 5), the conformation change according to this model led to less steric hindrance and thus less nanotubes dispersed (Figure 6). The increase in electrostatic repulsion as a stabilization mechanism was not enough to compensate for reduced steric hindrance, possibly because the lower acid character of the polymer investigated in this study. As the number of charges on the PAA chain increased, the PAA chain became less hydrophobic and therefore lost affinity for the nanotubes wall. Detaching during sonication thus became more likely, despite the strong MWCNT-affinity of the PS block. Although we had no direct evidence for the model in Figure 6, our data did not contradict this hypothesis.

According to our data, the affinity of the polymer for the nanotube was the limiting factor at low pH, whereas at high pH, the stabilizing ability of PAA decreased and became the limiting factor. Therefore, high styrene content led to the highest nanotube concentration at low pH and high PAA content led to the highest nanotube concentration at high pH.

### 3.4. Characterization via Zeta-Potential Measurements and SEM Microscopy

PS_26_PAA_580_, which was the polymer sample with the best nanotube stabilization efficiency at high pH, was analyzed in terms of zeta potential measurements (Table 2). Since a high absolute zeta potential value (>25 mV or <−25 mV) indicates good colloidal stability [32], the zeta potential at pH 5 was too small for stable colloidal dispersion, thus meaning that micelle encapsulation was indeed unlikely. Conversely, the zeta potential was high enough at pH 9.5 for the formation of stable colloids. In both cases, the incorporation of CNTs seemed to reduce the micelle stability of the system.

These findings suggest a shift in dispersion stabilization mechanism from steric hindrance to electrostatic repulsion. As shown by the zeta potential data (Table 2), PS_26_PAA_580_ was almost uncharged at pH 5, making it unable to stabilize nanotube dispersion by the electrostatic repulsion mechanism. Steric hindrance was in this environment the only stabilization mechanism. At high pH, the zeta potential was –58 mV, suggesting very good colloidal stability provided mainly by electrostatic repulsion as a stabilization mechanism. However, Figure 4 shows also that the amount of nanotubes dispersed decreased with pH raising. This could be explained by the increase in charge density in the PAA chain, which had two effects. Firstly, the PAA chain became less hydrophobic and therefore lost affinity for the nanotube wall. Detachment during sonication thus became more likely, despite the strong affinity of the PS block. Secondly, the increase in charge density on the PAA caused a conformational change (see Figure 6), thus severely reducing the steric hindrance. Notably, dispersions containing polymer with short PAA chain (PS_26_PAA_226_ and PS_26_PAA_81_) show a sharp decrease in absorbance (Figure 4), which was already found at pH 7. These short chains had less random walk steps and thus were already in rod shape conformation at pH 7. Longer chains could still make a highly stretched coil and therefore provided steric hindrance.

The nanotube dispersions were analyzed by SEM in both acidic and alkaline environments aimed at supporting the observations gathered from UV-Vis spectroscopy. In acidic conditions, the PS_26_PAA_226_ composite (Figure 7a) shows well separated CNTs structures thus suggesting their homogeneous distribution in the original dispersion before drying. In alkaline conditions, nanotubes were mostly aggregated (Figure 7b), which is a result of the conformational change of the polymer as hypothesized in Figure 6. Notably, in UV-Vis measurements, aggregated nanotubes were separated by centrifugation from the analyzed dispersion, thus explaining the low absorbance value for alkaline PS_26_PAA_226_ (Figure 3). Similarly, for PS_26_PAA_580_ a lower CNT concentration was found in solution when the pH was increased from 5 to 9.5. However, differently from PS_26_PAA_226_, the decrease was much less severe, because Figure 7e,f still shows nanotubes homogeneously dispersed suggesting good stability of the remaining nanotubes. The SEM images confirm the observations made with UV-Vis (Figure 3), because this also showed a milder response for longer PAA chains. In summary, the micrographs in Figure 7 were visual evidence of the pH-responsive stabilization of MWCNTs in water that was easily modulated by tuning the length of the PAA chain.

### 3.5. Resistivity of MWCNTs/Polymer Composites

The resistive behavior of the MWCNTs/polymer composites was finally evaluated by depositing water dispersions containing different MWCNTs content on an electrical circuit. The MWCNT content of the solid samples was estimated by using TGA (see Appendix A) by plotting the relative residue mass at 450 °C as a function of the alimentation content (Figure 8). A logarithmic empirical fit of the experimental data was used to estimate the actual MWCNTs weight percentage in the composites (Figure 8).

The PS_26_PAA_580_ copolymer was selected since it provided the best dispersions at high pH. After drying the MWCNTs/PS_26_PAA_580_ composite in an oven, the electrical resistance of the electrode was measured at room temperature and plotted (Figure 9) against the MWCNTs weight percentage calculated according to Figure 8. For alkaline samples, 1 M NaOH was added dropwise to the same dispersions until the desired pH was measured. This means that alkaline samples had the same CNTs concentration after drying. The electrical resistance was eventually measured on three replicates.

For conductive fillers in an insulating matrix, the conductance depends on the percolative networks among the nanotubes. The percolation threshold (critical filler content where resistance sharply decreases) [49] of a composite can be found by fitting the experimental data with Equation (1) [50].
(1)$$R\propto \frac{1}{{\left(\phi -\phi_{p}\right)}^{t}},$$
where *R* is the resistance of the composite, *Φ* is the filler content, *Φ_p_* is the filler content at the percolation threshold and *t* is the critical exponent, which is non-universal.

Figure 9 shows that alkaline samples display lower electrical resistance than the acidic ones, with a percolation threshold of approximately 8.2 wt% for the former and 9.3 wt% for the latter composites. This feature suggests that the composites obtained from alkaline dispersions possibly contained MWCNTs in closer proximity to each other, in agreement with microscopy investigations. Nevertheless, an effective contribution of the charge density on the electrical conductivity of MWCNTs dispersions cannot be neglected [51,52].

### 3.6. Investigation of pH Responsive Behavior of the Composite

These experiments well evidence the pH-responsive behavior of the MWCNTs/polymer composites. This behavior inspired us to carry out further studies aimed at determining the possible influence of the resistive character of the composite even in the solid state by means of an organic base dissolved in acetone, i.e., a non-solvent for the polymer. DABCO (pK_a_ is 8.82) was selected as the organic base since it is able to neutralize the acidic groups of the PAA block (pK_a_ of acrylic acid monomer is 4.25) of the PS_26_PAA_580_ copolymer. The electrodes with drop casted CNTs/PS_26_PAA_580_ sample were submerged in an acetone solution containing 5 g/L (0.45 mol/L) DABCO for half an hour. Indeed, a significant decrease in electrical resistance was found after immersion for 15 min (Figure 10), thus suggesting that the polymer microstructure could be altered even when deposited on a solid support. After removal from the acetone solution, the electrical resistance value suddenly spiked to very high values, possibly due to a quick drop in temperature, due to solvent evaporation. The effect of temperature on the resistivity was not surprising since CNTs are known to be sensitive to temperature variations [53,54].

The electrical resistance reached then a stable value only after 24 h out of the acetone solution. A possible explanation for this behavior was that water might have been removed from the composite by acetone. Since PAA is hygroscopic, the composite can slowly reabsorb water from atmospheric humidity. This suggests that absorbed water plays a role in the microstructure (and thus electrical resistance) of the composite. Notably, the humidity sensor made by using PAA has been effectively proposed in the literature by Wu et al. [55].

## 4. Conclusions

In this study, a series of pH responsive amphiphilic PS-b-PAA copolymers with a variable length of the PAA block were successfully synthesized. The polymers were found to be able to disperse MWCNTs directly by sonication in water and with a loading depending on their composition. UV-Vis and SEM investigations reported that PS-b-PAA copolymers with larger relative content of styrene units were able to better disperse carbon nanotubes in water, due to the chemical affinity between the aromatic moieties of the polymer and the graphitic nature of MWCNTs. Notably, the dispersion stability was also affected by pH, as evidenced by a change in absorption in UV-vis experiments and a decreased number of nanotubes visible from SEM micrographs. The stabilization ability of all polymers was higher at lower pH values possibly due to conformational changes of the PAA block, resulting in a different stabilization mechanism. At low pH, the stabilization mechanism is likely based on steric hindrance, since the zeta potential is too low for the alternative mechanism, electrostatic repulsion. Deprotonation of PAA at high pH caused improved electrostatic repulsion as evidenced by the higher zeta potential, but at the same time the charges on the polymer reduced the affinity for the nanotubes. Furthermore, the hypothesized globule to stretched conformational change parallel to the nanotube surface reduced steric hindrance, thus resulting in an overall decrease in CNTs stabilization.

Solid dispersions of the prepared mixtures resulted in being electrically conductive, with composites obtained from alkaline dispersion displaying lower percolation thresholds. This pH-dependent behavior was tentatively explained in terms of conformational changes of PAA from globule to stretched, which decreased the steric hindrance between the nanotubes and favored the formation of effective percolative networks. Moreover, by exposing the solid composite to an organic base dissolved in acetone, the resistance significantly dropped, thus suggesting the potential application in the field of sensing.

Overall, this paper evidenced the versatility of the prepared polymers in providing liquid or solid CNTs dispersions directly in water, with the pH response tuned by the block lengths of the amphiphilic polymer. This feature is merely illustrative, but it was designed to stimulate the exploration of novel possibilities to tailor and manage the electrical conductance of polymeric materials, having in mind possible applications where a pH-dependent electrical response is relevant, such as, for example, the design of sensors, wearable electronics or bio-inspired smart materials.

## Figures and Tables

**Figure 1 nanomaterials-09-01410-f001:**
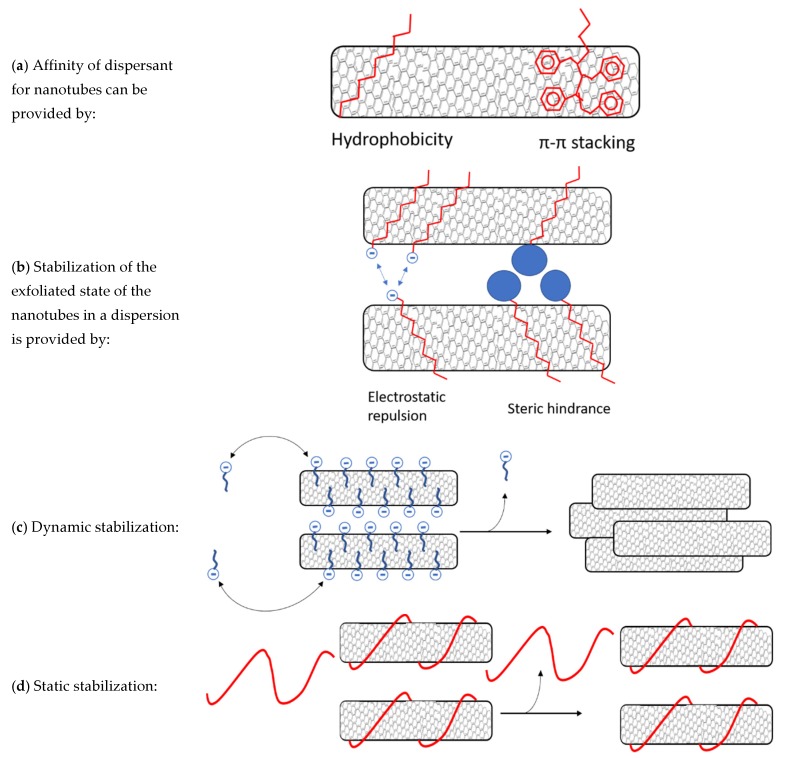
(**a**) Affinity of a dispersant for nanotubes can be provided by hydrophobicity, but even better by π-conjugated compounds that can stack on the nanotubes by π-π interactions. (**b**) There are two mechanisms that can prevent nanotube reaggregation. Firstly, electrostatic charges on the dispersant that repulse dispersants on other nanotubes. Secondly, bulky groups can hinder nanotubes from getting close to each other. (**c**) Surfactants and small molecules are stabilized dynamically. Dispersants can exchange easily from nanotube walls to the solvent. These dispersants are removed more easily resulting in reaggregation. (**d**) Larger polymers with high nanotube affinity can wrap around the nanotubes resulting in static stabilization where no exchange of dispersants takes place. These are harder to remove and can stabilize nanotubes in conditions like centrifugation, filtration, dialysis and precipitation.

**Figure 2 nanomaterials-09-01410-f002:**
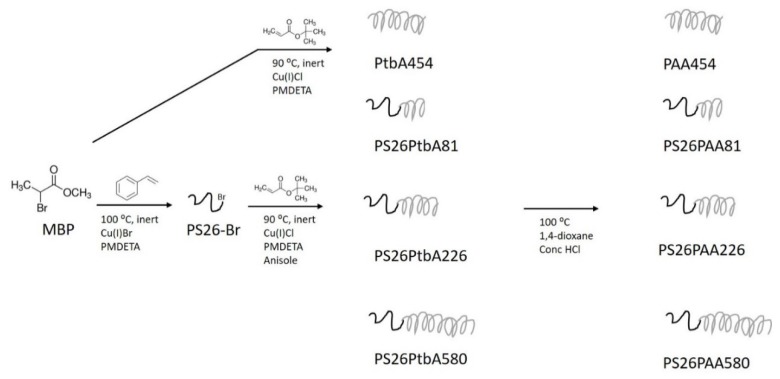
Process design for the synthesis of amphiphilic block copolymers used for the dispersion of multi-walled carbon nanotubes (MWCNTs). In the first step a polystyrene macroinitiator was made. Secondly, a chain of tert-butyl acrylate was formed in various lengths. The tBA groups were eventually hydrolyzed to form acrylic acid moieties.

**Figure 3 nanomaterials-09-01410-f003:**
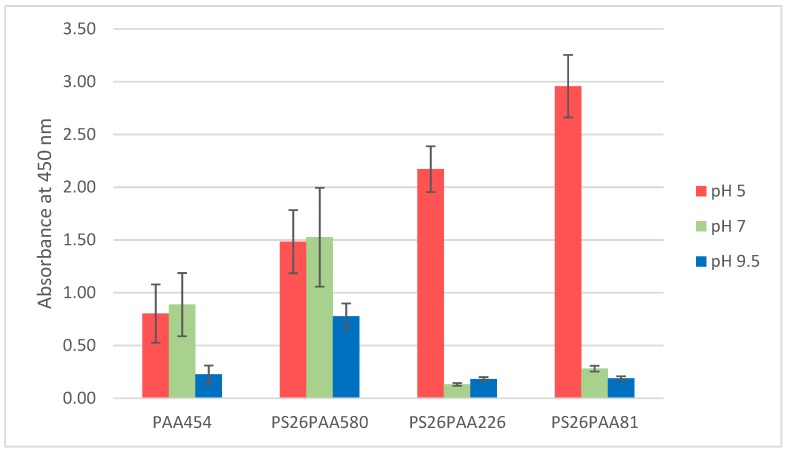
Absorbance values at 450 nm recorded from different polymer dispersions at three diverse pHs.

**Figure 4 nanomaterials-09-01410-f004:**
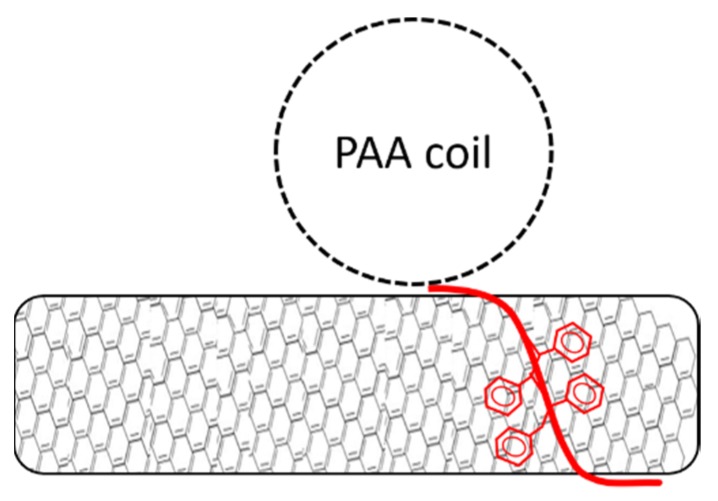
Schematic visualization of single chain adsorption on a nanotube surface.

**Figure 5 nanomaterials-09-01410-f005:**
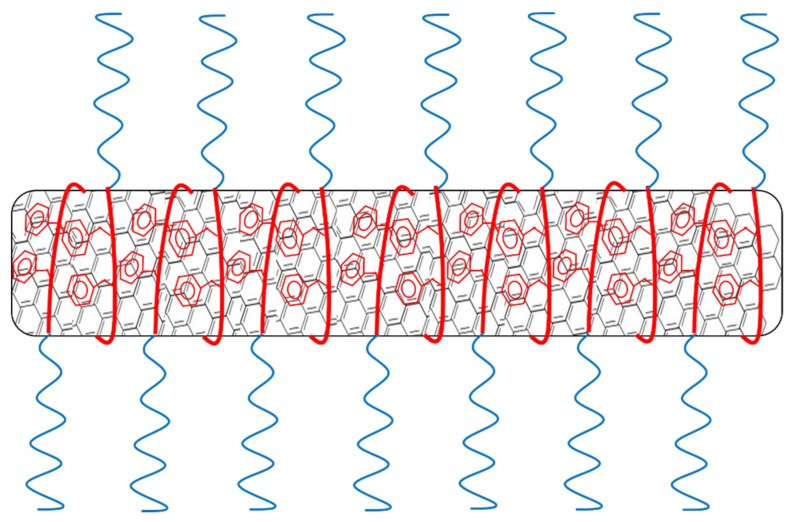
Schematic visualization of micelle encapsulation of nanotubes by the amphiphilic polymer.

**Figure 6 nanomaterials-09-01410-f006:**
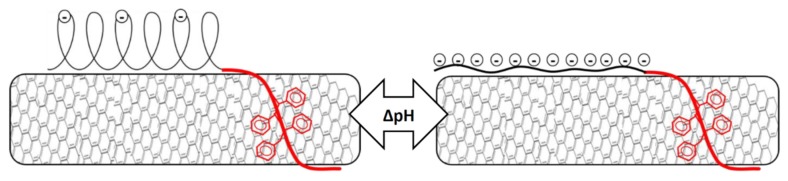
Schematic visualization of the consequences of pH change. The conformation of PAA changes from globular to rod-like resulting in reduced steric hindrance between the nanotubes.

**Figure 7 nanomaterials-09-01410-f007:**
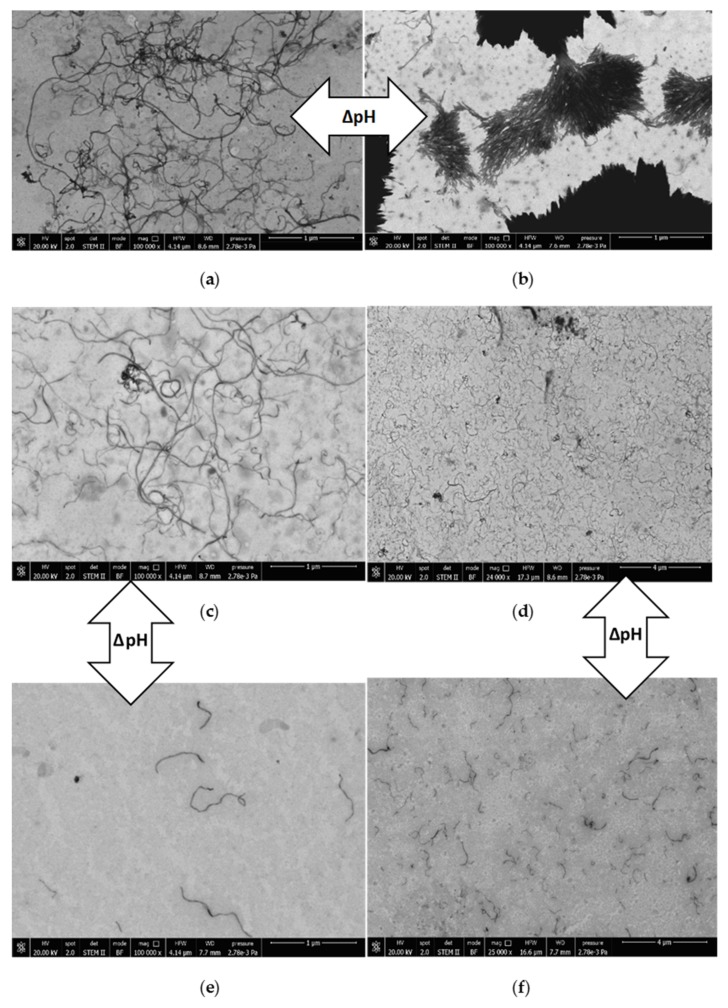
Field emission scanning electron micrographs of (**a**) PS_26_PAA_226_ pH 5, (**b**) PS_26_PAA_226_ pH 9.5, (**c**) and (**d**) PS_26_PAA_580_ pH 5, (**e**) and (**f**) PS_26_PAA_580_ pH 9.5.

**Figure 8 nanomaterials-09-01410-f008:**
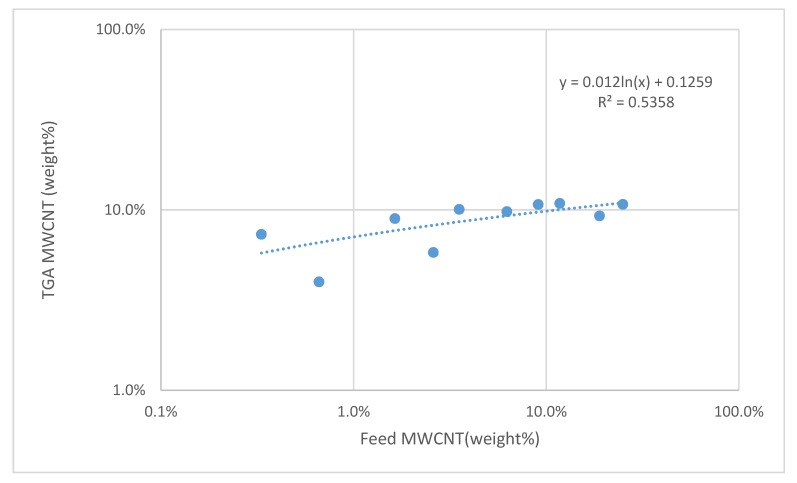
Plot of MWCNT wt% in the feed versus MWCNT wt% as obtained by TGA and fit with a logarithmic curve.

**Figure 9 nanomaterials-09-01410-f009:**
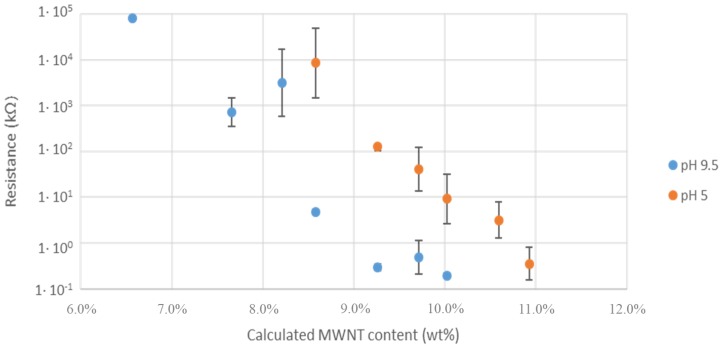
The electrical resistance of composites made from PS_26_PAA_580_ polymer. Increasing the wt% of MWCNTs results in reduced resistance. For every sample an acidic (pH 5) and alkaline (pH 9.5) composite was made.

**Figure 10 nanomaterials-09-01410-f010:**
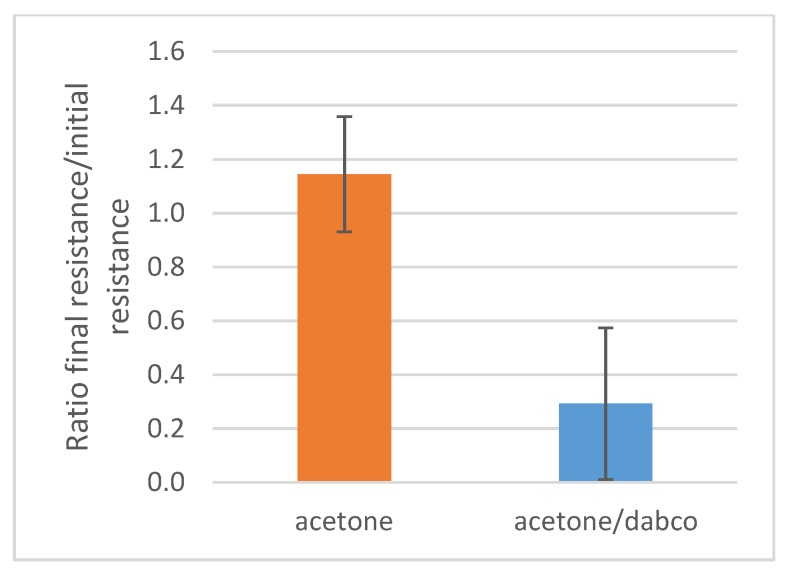
Composites dried on electrodes were submerged in acetone or in a DABCO solution in acetone (0.45 mol/L). After taking them out and waiting 24 h, the electrical resistance was compared with the original resistance from which a ratio was calculated. The average of four measurements was reported.

**Table 1 nanomaterials-09-01410-t001:** Total surface coverage of single polymer PAA chain for different polymers at 0.46 mg/mL concentration in water at pH 5.

Polymer	Concentration (mmol/mL)	Total Surface Coverage PAA (m^2^/mL)
PAA_454_	1.41·10^–5^	917
PS_26_PAA_81_	5.38·10^–5^	649
PS_26_PAA_226_	2.42·10^–5^	795
PS_26_PAA_580_	1.03·10^–5^	864

**Table 2 nanomaterials-09-01410-t002:** Zeta potential of polymer samples in water (6 mg/mL). MWCNTs were added for some samples with a feed of 0.22 mg/mL. Ten cycles were performed.

Sample	pH	Zeta Potential
PS_26_PAA_580_	5	−15 mV
PS_26_PAA_580_ with MWCNTs	5	−7 mV
PS_26_PAA_580_	9.5	−66 mV
PS_26_PAA_580_ with MWCNTs	9.5	−58 mV

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
