# Peer review of "Design of a pH-Responsive Conductive Nanocomposite Based on MWCNTs Stabilized in Water by Amphiphilic Block Copolymers"

_nanomaterials, 2019, doi:10.3390/nano9101410_

Round 1

Reviewer 1 Report

The manuscript entitled “Design of a pH-responsive conductive nanocomposite based on MWCNTs stabilized in water by amphiphilic block copolymers” can be accepted only after major revision

Please see below the comments

The manuscript should be corrected for grammar and the abstract needs revision. Section 3.1 can be divided into subsection for a clear discussion on pH variations. The language can be made simpler and the results and discussion can be sub sectioned for better clarity to the reader. Cite the latest references for the comparison of native stability of the amphiphilic block copolymers. The figure caption statement should be rewritten for TGA. Supplementary files also require correction and the sections should be highlighted with marking subsection. The exact relation between f and pH for a PAA chain depends on the polymer structure and length. Elaborate

Author Response

point by point response to reviewer 1

The manuscript should be corrected for grammar and the abstract needs 
revision.

R: A grammar check has been made and mistakes corrected, the abstract has been revised

Section 3.1 can be divided into subsection for a clear discussion on pH variations. The language can be made simpler and the results and discussion can be sub sectioned for better clarity to the 
reader.

R: We divided now section 3 in 6 subsections (instead of the previous 2) and simplified the discussion. We thank the reviewer for the suggestion, we think that now the text is easier to follow.

Cite the latest references for the comparison of native  stability of the amphiphilic block copolymers.

R: We couldn’t find studies about the native stability of amphiphilic block copolymers in the recent literature. To support our discussion, we now included some observations made during our study about decrease in viscosity upon sonication of polymers solution, which suggests disruption of micellar aggregates.

The figure caption  statement should be rewritten for TGA.

R: The caption has been rewritten

Supplementary files also require  correction and the sections should be highlighted with marking subsection.

R: The supplementary file has been revised, improved and divided in sections, referring to the corresponding sections in the main text.

The exact relation between f and pH for a PAA chain depends on the polymer structure and length. Elaborate"

R: The text in the supporting information has been revised to make more clear this statement

Reviewer 2 Report

The authors need to provide TEM images of the nanocomposites to provide the matrix morphology. In Figure 3: Please indicate the significance of this figure in the manuscript. In line 194-195: “…the CNTs dispersed by ultrasonication for 5 minutes…” How did the authors optimize this time duration? The authors may study the absorbance change with time for optimum sonication time. The authors should make a new subheading and place the polymer synthesis part discussions along with figure 2. The polymer preparation (section 2.2) is the most important part in this work. The authors need to clarify few part: How did the authors determine the reaction end time (as mentioned 5 hrs in line 119) as it varies batch to batch in addition polymerization reaction. Please indicate the abbreviations where it introduced first (MBP, PMDETA etc). In second paragraph of section 2.2: “To prepare the second block of the…” please mention the name of the second block there. In line 133: What is the concentration of 1-4 dioxane solution?

Author Response

point by point response to reviewer 2

The authors need to provide TEM images of the nanocomposites to provide the matrix morphology.

R: we think that for our purposes SEM investigations provide better insights for the determination of phase dispersion behavior of MWCNTs within the polymer matrix being the aggregates that generated after chemical solicitations in the micro-submicro-scale.

In Figure 3: Please indicate the significance of this figure in the manuscript.

R: Figure 3 shows a typical UV adsorption spectra of MWCNT dispersed in water. To shorten the long discussion, we moved it in the supporting information

In line 194-195: “…the CNTs dispersed by ultrasonication for 5 minutes…” How did the authors optimize this time duration? The authors may study the absorbance change with time for optimum sonication time.

R: The reviewer is right. We actually made in the past an extensive study concerning the effect of ultrasonication on the exfoliation of the same MWCNTs used in this work. In the paper European Polymer Journal 49 (2013) 1471–1478, we demonstrated that 5 min of ultrasonication were enough to exfoliate the graphitic bundles without affecting the aspect ratio of the single carbon nanotube. Very recently, Absorbances studies have been carried out with similar systems and water dispersions and confirmed the linear correlation between the amount of light absorbed (i.e., scattered) and the MWCNTs content in the solvent dispersion as also recently reported . We modified the text as the following:

“The copolymers were dissolved in water (0.46 mg/mL) and the CNTs dispersed by ultrasonication for 5 minutes (0.03 mg/mL feed) being this time interval effective in nanotubes deboundling without affecting their aspect ratio [add references: a) European Polymer Journal 49 (2013) 1471–1478; b) ref. 14 of this work].

The authors should make a new subheading and place the polymer synthesis part discussions along with figure 2.

R: Also based on another reviewer comments, section 3 is been subdivided in more subsections (Figure 2 belongs to the section describing the synthesis

The polymer preparation (section 2.2) is the most important part in this work. The authors need to clarify few part: How did the authors determine the reaction end time (as mentioned 5 hrs in line 119) as it varies batch to batch in addition polymerization reaction.

R: As specified in the introduction, the synthesis is based on a procedure described in our previous research (reference 36). The reference is also added now to experimental part and results and discussion for clarity.

Please indicate the abbreviations where it introduced first (MBP, PMDETA etc). In second paragraph of section 2.2: “To prepare the second block of the…” please mention the name of the second block there. In line 133: What is the concentration of 1-4 dioxane solution?

R: These points have been revised (see track changes in the manuscript).

Reviewer 3 Report

The paper has been largely improved over the previous versions. I recommend publication

Round 2

Reviewer 1 Report

The manuscript can now be accepted after minor changes in English and methodologies.  

Author Response

We further improved and corrected the English and rewrote part of the experimental section concerning the methodologies used to make it more readable.